# Hepatic Enzyme Profile in Horses

**DOI:** 10.3390/ani12070861

**Published:** 2022-03-29

**Authors:** Katy Satué, Laura Miguel-Pastor, Deborah Chicharro, Juan Carlos Gardón

**Affiliations:** 1Department of Animal Medicine and Surgery, Faculty of Veterinary, CEU-Cardenal Herrera University, 46115 Valencia, Spain; laura.miguel@uchceu.es (L.M.-P.); debora.chicharro@uchceu.es (D.C.); 2Department of Animal Medicine and Surgery, Faculty of Veterinary and Experimental Science, Catholic University of Valencia-San Vicente Mártir, 46001 Valencia, Spain; jc.gardon@ucv.es

**Keywords:** hepatobiliary disease, liver, enzyme profile, horse

## Abstract

**Simple Summary:**

This review aims to highlight the importance of using enzyme profiles for the diagnosis of common liver disorders in the horse. This review also highlights the limitations and alternative explanations for isolated or collective abnormalities of the different liver enzymes. This review also provides information on the screening test(s) of choice, their interpretation, and the results to confirm the diagnosis. Similarly, documented liver cases in the equine clinic are reviewed, with emphasis on abnormal enzyme activity.

**Abstract:**

For diagnostic purposes, liver enzymes are usually classified into hepatocellular and cholestatic. These two groups of equine liver-specific enzymes include sorbitol dehydrogenase (SDH), glutamate dehydrogenase (GLDH), γ-glutamyl transferase (GGT), aspartate aminotransferase (AST), lactate dehydrogenase (LDH), and alkaline phosphatase (ALP). SDH and GLDH mostly reflect hepatocellular injury and cholestasis, while GGT expresses high values in biliary necrosis or hyperplasia. Likewise, AST, LDH, and ALP also reflect hepatocellular and biliary disease, but these enzymes are not liver specific. From the clinical point of view of the course of liver or biliary disease, AST and ALP are indicative of chronic disease, whereas SDH, GGT, and GLDH indicate an acute course. The patterns of enzymatic changes at the blood level are associated with different types of liver pathologies (infectious, inflammatory, metabolic, toxic, etc.). Increases in hepatocellular versus biliary enzyme activities are indicative of a particular process. There are different ways to diagnose alterations at the hepatic level. These include the evaluation of abnormalities in the predominant pattern of hepatocellular versus cholestatic enzyme abnormalities, the mild, moderate, or marked (5–10-fold or >10-fold) increase in enzyme abnormality concerning the upper limit of the reference range, the evolution over time (increase or decrease) and the course of the abnormality (acute or chronic).

## 1. Introduction

Within the concept of liver disease, we can include different pathologies that directly or indirectly affect liver function. In turn, these alterations in liver function may be temporary or may progress to complete and irreversible failure [1]. Although liver failure is rare in equines, different authors point out that the clinical manifestation occurs when more than 70% of the function of this organ is lost [1,2,3,4]. Liver disease and liver failure are two distinct and important concepts that must be differentiated. Among all liver functions, the reserve is extremely important. Therefore, a loss of at least 60–80% of the mass of this organ is required to show clear signs of liver failure. This clarification becomes important because some indicators of liver function, such as albumin or coagulation times, are affected when this organ is insufficient and are therefore not sensitive for detecting disease in early stages [2,3,5].

Several different etiological causes may be ascribed to the development of horse hepatic diseases [1]. Pre-existing sepsis, hypoxia, neoplasia, toxic or metabolic causes in both foals and adult horses can lead to liver damage [1,6,7,8]. The consumption of toxic plants (e.g., mugwort and clover) and mycotoxins (e.g., aflatoxin, zearalenone, and fumonisin) also can develop hepatic disease [9,10,11,12].

Infectious and non-infectious causes of liver and biliary tract inflammation have also been described. These include serum hepatitis or Theiler’s disease or viral hepatitis (caused by equine herpesvirus, hepatitis, equine hepacivirus, and equine parvovirus), parasitic hepatitis (caused by large strongyles and ascarids), Tyzzer’s disease (*Clostridium piliforme*), inflammatory diseases such as cholangiohepatitis (due to cholelithiasis or intestinal obstruction), displacement of the right dorsal colon with bile duct obstruction, cholelithiasis, hepatic torsion, portal vein thrombosis and hyperlipemia [8,13,14,15,16,17]. Other causes of liver disease are primary neoplasms such as hepatocellular carcinoma, cholangiocarcinoma, or hepatoblastoma. Additionally, metastatic dissemination of lymphomas or malignant melanomas to the liver from other primary locations [8,18]. The main analytes to evaluate in blood samples related to liver disease are serum enzymes and indicators of residual liver functional capacity [6]. In general terms, we can classify liver enzymes into two main groups. On the one hand, some are filtered from the cytoplasm of the damaged hepatocyte called hepatocellular, and on the other hand, those that increase their concentration in the blood due to an increased synthesis as a response to the decrease or absence of normal bile flow from the liver to the duodenum also called cholestasis. In hepatocellular or cholestatic forms of liver injury, these liver enzymes are released into the bloodstream and, therefore, increased serum levels are diagnostically useful [19]. Therefore, the so-called liver functionality tests measure the serum level of liver enzymes and thus reflect hepatocyte integrity or cholestasis rather than the liver function itself [3]. Thus, in horses with liver disease, serum enzyme levels are related to the concentration of the enzyme in the hepatocyte, the severity, and duration of the disease, and the half-life of the enzyme. In general, the duration of elevation of serum liver enzyme activity depends on molecular size, intracellular location, plasma elimination rate, enzyme inactivation, and increased hepatic production. [19]. This results in a measurable increase in enzyme activity in the serum of those animals with hepatocellular injury [8].

## 2. Types and Characteristics of Hepatic Enzymes in Horses

Standard biochemical indices of hepatocellular disease include sorbitol dehydrogenase (SDH), glutamate dehydrogenase (GLDH), aspartate aminotransferase (AST), and lactate dehydrogenase (LDH) while indicators of hepatobiliary disease include γ-glutamyl transferase (GGT). Alkaline phosphatase (ALP) indicates both hepatocellular and biliary origin [3,20]. These enzymes can usually be found in the cytoplasm (AST and LDH), mitochondria (GLDH and AST), nucleus or membranes (ALP and GGT) of hepatocyte cells, where they catalyze specific reactions [20] (Table 1).

Cytoplasmic enzymes are released at the beginning of cell degeneration, whereas mitochondrial enzymes are released after advanced cell necrosis [20]. Based on this, GLDH, AST, and LDH activity are increased in hepatocellular damage while GGT and ALP enzymes are increased in cholestatic liver disease in horses [4,21]. Based on their specificity, liver-specific enzymes include SDH, GLDH, and GGT, where, on the one hand, SDH and GLDH reflect hepatocellular damage and GGT, on the other hand, is indicative of biliary damage. Other enzymes such as AST and LDH also reflect hepatocellular disease while ALP indicates biliary damage. However, these last three enzymes are not specific to the equine liver [2,3] (Table 2).

These parameters are estimated values based on a review of available reports [2,3,20,22,23].

SDH is a liver-specific enzyme in horses. The reference range of SDH in equines is 0–8 IU/L [24]. A drawback is the instability of this enzyme in serum or plasma, even when refrigerated or frozen. According to Fouche et al. [25], plasma SDH activity in refrigerated samples at 4 °C is suitable for analysis for 24 h. It is not recommended to store plasma for more than 4 h at room temperature or −20 °C. This enzyme has a half-life of 12 h, and after an acute event, baseline values can be observed 3–5 days later [8].

The GLDH enzyme is found in the mitochondria of hepatocytes, and horses with an acute hepatocellular disease and its blood levels are abnormally high. The range of reference plasma values for GLDH in the equine species is <3.5 IU/L [24]. GLDH is more stable compared to other enzymes and has a somewhat longer half-life than SDH and, because of its stability, is a recommended test for detecting acute hepatocellular disease [2,3,23]. However, this parameter should be interpreted with caution in foals, as GLDH levels are usually increased in young lactating foals without actual liver disease [19].

AST is present in the cytoplasm of liver cell mitochondria. The reference range of AST in equines is 150–270 IU/L [24]. Serum levels are usually increased in liver disease and reflect hepatocellular injury. However, since it is also found in the liver, heart, skeletal muscle, and kidney, elevated serum activity can only be interpreted in conjunction with other more specific liver enzymes to diagnose disease. It is also important to note that this enzyme can be increased by hemolysis, as it is present in erythrocytes, and by lipemia [5].

GGT is an enzyme widely distributed in various tissues. Other organs such as the lungs, kidneys, pancreas, and mammary glands also produce GGT. However, these amounts are small, so this enzyme is considered liver specific. However, the activity of this enzyme in serum or plasma originates almost exclusively from hepatocytes [19]. This fact confers to GGT a high specificity for diseases of the hepatobiliary system and its reference range in equines is 5–20 IU/L [24]. Increased GGT activity can be considered normal in foals, donkeys, and mules. In them, serum levels can increase up to 3 times the normal reference range for horses. In foals, during the first month of life, values were 1.5 to 3 times higher than the upper physiological reference values for healthy adult horses. In neonatal foals, serum levels increase during the first two weeks of life because GGT concentrations are higher in colostrum and milk [26].

ALP reflects biliary injury, but is not specific to the liver, as it is also produced in bone, intestine, and macrophages. Care must be taken with the interpretation of this enzyme in growing animals, where normal values are 2 to 3 times higher than reference values in adults due to increased bone turnover associated with physical growth [24]. The reference range of ALP in equines is 73–194 IU/L [24]. In equids, ALP activity is used as a test of liver excretory function. ALP increases after 48 h of liver damage and is usually higher in cholestasis than in hepatocellular damage [24].

LDH isoenzyme 5 (LDH-5) is a non-specific enzyme as it is abundant in the liver, although it is also present in kidneys, muscle, myocardium, and red blood cells [5]. The reference range of LDH in equines is 162–412 IU/L [24].

Based on the clinical point of view of the course of liver or biliary disease, AST and ALP are indicative of chronic disease, whereas SDH, GGT, and GLDH indicate an acute course [3]. The initial activity of an enzyme in plasma or serum is usually a reflection of the amount and turnover of the tissue containing this enzyme. Thus, serum concentrations of specific liver enzymes are usually higher in acute liver disease than in chronic liver disease. Increased activities of SDH, GLDH, and AST occur even with mild acute hepatocellular injury, and the magnitude of the enzyme increase may not correspond to the functional state of the liver [3]. However, in chronic liver diseases with severe fibrosis and a reduction in the number of functional hepatocytes, plasma/serum liver enzyme activities may be within normal limits [4,21].

## 3. Assessment of Liver Enzyme Abnormalities

The increase in one or more enzymes is usually expressed as increments of levels above the upper limit of the reference interval. Usually, a 2- to 3-fold increase above the reference range is considered mild, while a 4- to 5-fold increase is moderate, and when the value approaches or exceeds a 10-fold increase, it is considered marked (Table 3). The degree of the increase in hepatocellular-damage, enzyme activities may help stage disease severity [27].

Relative increases in hepatocellular versus biliary enzyme activities may point the clinician to a particular process (Table 4). Thus, if GGT activity is greatly increased and GLDH, SDH, or AST activity is only moderately increased, a process involving the biliary system, such as cholangiohepatitis, is most likely to be suspected. Conversely, if hepatocyte enzyme activity is very high and GGT activity is slightly increased, then the disease predominantly involves hepatocytes, such as hepatitis [19,20]. Other causes of liver disease may result in a similar increase in hepatocellular and biliary enzyme activities, such as pyrrolizidine alkaloid toxicity and hepatic lipidosis. These processes are probably highly dependent on the duration of the disease. Thus, if serum enzyme activity decreases by 50% for 2–4 days, it suggests that damage has ceased. Conversely, if the activity remains constant over time or increases, it suggests that damage is continuing.

## 4. Hepatic Enzymes Related to Hepatic Disease

### 4.1. GGT

GGT is an excellent marker of liver disease in the horse [23]. The increase in GGT activity is very specific to liver disease, as pathologies in organs in which it is also expressed, such as the kidney and pancreas, do not induce changes in this enzyme in the blood. In the horse with moderate or severe liver disease, increased serum GGT may be associated with hepatocellular damage and hepatic necrosis. In this regard, a retrospective study of 50 horses with liver disease showed increased GGT in all cases [23]. In cases of acute liver disease, levels of this enzyme may remain elevated for several weeks or even continue to rise despite the animal’s apparent clinical improvement [8]. Similarly, after the resolution of acute liver injury, GGT activity may continue to increase for several days, probably as a result of biliary hyperplasia [3]. Foals with pneumonia caused by *Rhodococcus equi* have elevations of GGT and other enzymes that should be properly evaluated for possible causes of intra-abdominal abscesses. However, although the greatest increase in GGT activity is seen with biliary disease, it should also be considered that this enzyme may be released in small amounts as a consequence of hepatocellular injury [19]. On the other hand, if it is the case that several horses housed together show increased GGT activity, possible intoxication should be taken into account. In fact, GGT activity may also indicate subclinical exposure to hepatotoxins in outbreaks of pyrrolizidine alkaloid toxicity [10,11]. Increases in GGT activity are usually related to inflammation, biliary hyperplasia or destruction of the epithelium of the bile canaliculi, and secondary inflammation of cholestatic hepatocytes [3]. Large increases in cases of biliary obstruction, with values up to 1670 IU/L, have been founded [28]. Other diseases are also associated with increased GGT, such as bile duct ligation, cholestasis, iron, or copper hepatotoxicosis, and hyperlipidemia [8,29]. In equine cholestasis, GGT increases 9-fold while ALP increases 2-fold [30]. In addition, DeNotta et al. [22] reported that serial GGT measurements can predict the duration of antimicrobial treatment in cholangiohepatitis of bacterial origin. In contrast, cases of chronic liver fibrosis are the only disease in which an abnormal increase in GGT might not be observed. In horses with dorsal displacement of the right colon or proximal small bowel disease, bile flow obstruction usually occurs. Increased GGT enzyme levels may also be observed in these cases [26]. Thus, elevated serum GGT levels become a method with high levels of sensitivity, for the detection of liver damage secondary to proximal enteritis in equines [31]. This is also a routine practical test for the evaluation of hepatic amyloidosis status [8]. On the other hand, elevated levels of both GGT and SDH associated with sepsis, perinatal asphyxia syndrome, or proximal enteritis have been reported in a study performed in hospitalized newborn foals [32].

In ponies receiving 200% of their maintenance metabolizable energy requirement for 2 years, Schedlbauer et al. [33] found a 6.2-fold increase in ALP and a 2.3-fold increase in GGT over normal values. Serum GLDH activities in ponies were 2.4 times higher than the upper reference range, which is normally 8.9 U/L [34]. Since changes in serum AST and GLDH activity have been higher in ponies than in horses, it is speculated that the liver of ponies was more affected by the effect of early obesity compared to horses. Serum GGT activity of ponies with laminitis exceeded the reference range at the end of the study. Accordingly, Chameroy et al. [35] observed elevated serum GGT activities in 64.3% of obese horses with a history of laminitis.

Whereas GGT is a less specific marker of liver alteration in humans, it has been shown that high levels of GGT can be an independent predictor of the incidence of type 2 diabetes [36]. In horses, type 2 diabetes mellitus is often associated with pituitary pars intermedia dysfunction (PPID) [37]. Indeed, different alterations in lipid metabolism such as lipolysis or ketogenesis have been detailed in horses with PPID [38]. In ponies, cases of persistent hypertriglycemia, hyperinsulinemia, and hyperglycemia attributed to PPID have been reported, as well as an increase in the activity of different liver enzymes such as ALP with values up to 204 IU/L, AST (693 IU/L), GGT (46 IU/L) and SDH (10 IU/L). Likewise, liver biopsies have shown important changes in the vacuoles of liver cells corresponding to hepatic lipidosis [39].

In racehorses, a moderate increase in GGT activity (50 to 140 IU/L) has been observed with little or no increase in the activity of other liver enzymes, including ALP [40]. GGT activity has been shown to correlate with cumulative training load and running frequency [40,41]. In this regard, oxidative stress has been hypothesized as a cause [3]. More recently, Ramsay et al. [42] revealed increased serum SDH and GGT activity in Thoroughbred racehorses. Depletion and repeated replenishment of hepatic glycogen stores during intensive training and/or hepatocellular injury have been proposed as possible causes [40]. In fact, SDH is a more specific marker of hepatocellular damage than GGT [43]; therefore, a frequent increase in SDH suggests that the observed elevations in GGT may also have a hepatic origin. Considering that the half-life of SDH is much shorter than that of GGT, asynchronous increases in the activity of both enzymes are not always synonymous with liver disease [8,13]. Furthermore, it has also been reported that, when serum levels are ≥100 IU/L, they are generally associated with poor animal performance [44].

### 4.2. SDH

SDH is a good indicator of the active hepatocellular disease, which is increased in moderate and severe cases of intrahepatic and extrahepatic cholestasis [5]. Although hepatocellular damage can be a sign of primary liver disease, it is important to remember that systemic hypoperfusion, anemia, and side effects of gastrointestinal pathology can also cause hepatocellular damage in the absence of primary liver pathology. Significant increases in hepatobiliary enzymes occur in cases of neonatal sepsis. In cases of severe liver failure with massive increases in SDH and GGT enzymes, hyperbilirubinemia, hypoalbuminemia, hypoalbuminemia, hypoglycemia, and neurological signs should raise suspicion of Tyzzer’s disease [45]. When hepatocellular damage is present, there is a sharp rise in serum SDH, followed by a marked decline because of its short half-life. Therefore, repeated SDH measurements may be useful for determining the resolution or progression of the acute hepatocellular disease [3] but are less appropriate for the diagnosis of chronic liver disease.

### 4.3. GLDH

GLDH in elevated serum concentrations is considered highly specific for liver disease. These may include primary liver tumors, metastatic infiltration of the liver, or hepatic lipidosis. In addition, certain viral diseases such as equine parvovirus, non-primitive hepacivirus, or equine pegivirus, have significantly elevated levels of this enzyme. In experimental equine parvovirus infection, the increase in GLDH activity precedes (2–7 days) and is of greater magnitude (versus the upper reference limit) than the increase in SDH activity, and in some animals, GLDH activity remains elevated for 2–10 days after SDH activity normalizes [17]. Although GLDH reaches values above 600 U/L in the horse, in most animals, the activity of this enzyme does not exceed <200 U/L. This increase in enzyme activities responds to hepatocyte necrosis secondary to viral infection and is maintained for 6 and 10 weeks. The apparent higher sensitivity of GLDH versus SDH for parvovirus hepatitis may be explained by the inflammation identified in the liver, which is mainly in the centrilobular region [17].

GLDH activity increases with liver damage in response to ischemia and liver toxicity [4,20]. Intestinal disorders can lead to secondary liver pathology through ascending biliary tract infection, exposure of the liver to endotoxin and bacteria carried by the portal circulation, and secondary to hypoperfusion associated with shock and sepsis. Hepatocytes are also susceptible to hypoxic injury associated with anemia, which is generally reflected in an increase in hepatocellular injury enzymes such as SDH, AST, and GGT [31]. However, serum GLDH activity is highly variable in ponies exposed to pyrrolizidine alkaloids, suggesting that this enzyme is only useful for the diagnosis of acute liver injury [46].

The sensitivity of GLDH activity for the detection of liver necrosis, lipidosis, and cirrhosis is 78%, 86%, and 44%, respectively [47]. Indeed, horses with severe chronic fibrosis (cirrhosis) may occasionally have SDH and GLDH activities within normal reference ranges [4]. They may be within normal limits in the later stages of subacute or chronic liver disease [48]. On the other hand, in horses anesthetized with halothane, approximately 12-fold increases in GLDH activity and 2-fold increases in AST activity were observed 24 h after anesthesia. This may also reflect the effect of reduced oxygen tension on the centrilobular localization of GLDH activity [49].

In a two-year, on-farm investigation of forage-associated liver injury, weanling foals, yearlings, and mature horses all showed an increase of up to 1000 IU/L in GGT concentrations and up to 1200 IU/L in GLDH. However, no animal showed signs of liver failure, and total bilirubin and bile acid concentrations remained within the range of reference values in almost all horses. These studies demonstrate that the size of changes in serum hepatocellular enzyme levels does not determine the prognosis of the disease. Therefore, in horses liver disease is best determined by integrating abnormalities such as function tests, etiology, degree of fibrosis on liver biopsy, or brain dysfunction caused by the inability of the liver to eliminate toxins [4]. In similarity to SDH, GLDH has been shown to be non-specific in determining the cause of liver injury.

### 4.4. AST

AST is increased in reversible and irreversible hepatocyte injury following hepatocellular injury and cholestasis. Hepatic necrosis due to viral hepatitis, acute biliary obstruction, cholangiohepatitis, among others, can cause significant increases in this enzyme. The sensitivity of serum AST activity in horses is 72% for hepatic necrosis and 100% for hepatic lipidosis [2]. In addition, serum AST increases after myocyte injury. As serum AST activity does not differentiate between hepatocellular or myocyte injury, additional testing using liver-specific (SDH) or muscle-specific enzymes, such as creatine kinase (CK), is required. A marked increase in serum AST and SDH suggests acute or active hepatocellular injury, and a marked increase in serum AST with modest to moderate SDH activity suggests chronic liver injury or recovery from acute liver injury. An increase in serum AST along with CK is a clear indication of muscle damage [47].

The increases in serum AST and serum SDH or GLDH suggest acute or active hepatocellular injury. However, marked increases in serum AST with mild or moderate increases in SDH or GLDH suggest either chronic liver injury or recovery from an acute injury [30].

### 4.5. ALP

GGT and ALP are the main indicators of post-hepatic cholestasis [3,19]. However, serum GGT is more sensitive than ALP for the diagnosis of these processes [19]. AST, GGT, and ALP are also increased in acute gastrointestinal disorders. A retrospective study in colicky horses revealed increased GGT activity in 49% of horses with right dorsal colonic shift, and in only 2% of horses with a left dorsal shift. This finding was attributed to extrahepatic biliary obstruction due to compression of the bile duct by the displaced colon [26]. In horses with proximal enteritis related to liver injury secondary to enteric bacteria ascending from the common bile duct, endotoxin absorption from the portal circulation and/or hypoxia, AST, GGT, and ALP enzyme activities are markedly increased [31]. The increased enzyme activity in these cases is possibly related to the anatomical proximity between the intestine and liver and direct communication through the biliary system and portal circulation [22]. Some pathologies such as Theiler’s disease, liver lobe torsion, and plant toxicity have elevated levels of this enzyme [3].

### 4.6. LDH

LDH is poorly specific to liver damage because it also is released from muscle. Unlike AST, LDH has a short half-life (<24 h). High values for SDH, AST, and LDH do not indicate specific liver disease, although elevations are more likely with hepatocellular conditions. However, to differentiate liver disease from muscle cell injury, LDH-5 isoenzyme can be measured [20]. Elevated levels of this enzyme can be found in diseases produced by iron, copper, or mycotoxin intoxications.

Table 5 list the different enzyme profiles documented for several liver pathologies in horses:

## 5. Conclusions

The liver enzymes present in serum reflect, to different degrees, different pathologies such as obstruction, proliferation, inflammation, toxicity, or neoplasia of both the liver and the hepatobiliary duct system. Their use is oriented to two purposes. The first is to evaluate and establish the type and degree of injury and the second is to provide treatment and prognosis for diseases that alter liver function. Depending on the evaluation, any of the enzymes involved offers data that are important to know how to interpret to reach a proper diagnosis. In general, variations in enzyme profiles are associated with different types of liver pathologies. Thus, acute or severe and preferentially hepatocellular injury is primarily associated with elevated levels of SDH and GLDH enzymes and increased GGT values. However, persistent injury is associated with weak changes in hepatocellular and hepatobiliary enzymes and thus other diagnostics are required to define the cause of these consistent increases. Thus, persistent acute or chronic liver damage is likely to disrupt one or more liver function tests. Additionally, when GGT and ALP enzymes are increased, they have important diagnostic value in determining the presence of decreased or disrupted bile flow. Therefore, it is easy for clinicians to differentiate and correctly diagnose liver pathologies from other diseases by evaluating a single hepatobiliary enzyme. The correlation of the variation of several enzymes together with other biochemical markers allows increasing the sensitivity and specificity to identify liver diseases to obtain an adequate treatment and prognosis.

## Figures and Tables

**Table 1 animals-12-00861-t001:** Characteristics of hepatic enzymes in horses based on location and function.

Origin	Type	Location	Function (Catalysis)
Hepatocellular	SDH	Cytoplasmatic	Conversion of fructose to sorbitol
	GLDH	Mitochondrial, in the centrilobular areas of the liver	Conversion of glutamate to 2-oxoglutarate
	AST	Cytoplasmatic of hepatocytes and other tissues, including skeletal muscle	Conversion of aspartate and alpha-ketoglutarate to oxaloacetate and glutamate
	LDH-5	Cytoplasmatic	Reversible transformation of pyruvate to lactate
Biliary	GGT	Microsomal membranes in the biliary epithelium, and also in the canalicular surfaces of hepatocytes <5% is found in the cytoplasm	Cleaves C-terminal glutamyl groups from amino acids and transfers them to another peptide or amino acid. GGT is important in glutathione metabolism (reduced and oxidized GSH are the main targets) and amino acid absorption (cysteine in the kidney)
	ALP	Epithelium of biliary canalicular membrane, and sinusoidal membrane of hepatocytes	Non-specific metalloenzyme which hydrolyzes many types of phosphate esters at an alkaline pH in the presence of zinc and magnesium ion

SDH: sorbitol dehydrogenase; GLDH: glutamate dehydrogenase, AST: aspartate aminotransferase; lactate dehydrogenase (LDH); γ-glutamyl transferase (GGT); alkaline phosphatase (ALP).

**Table 2 animals-12-00861-t002:** Reference value interval, half-life, sensitivity, specificity, and stability of the hepatobiliary enzymes in horses.

Enzyme	Reference Value Interval (UI/l)	Half-Life	Sensitivity	Specificity	Stability
SDH	2–8	<12 h	+++	++++	+
GLDH	2–10 In foals, GLDH increase is compared to adult	12–24 h	+++	+++	++
AST	150–300	7–8 days	+++	+	++++
GGT	5–20	3 days	++++	+++	++++
ALP	120–250 Foal and growing: 100-fold greater than in adults	3 days	++	+	++++

SDH: sorbitol dehydrogenase; GLDH: glutamate dehydrogenase, AST: aspartate aminotransferase; γ-glutamyl transferase (GGT); alkaline phosphatase (ALP) lowest (+); highest (++++).

**Table 3 animals-12-00861-t003:** Assessment of liver enzyme abnormalities.

Increase	Number of Times Higher than the Upper Reference Interval
Mild	<5 times
Moderate	5–10 times
Marked	>10 times

**Table 4 animals-12-00861-t004:** Patters of alterations of hepatic enzymes.

Number of Times Higher than the Upper Reference Interval	Pattern
>10-fold increase in SDH, GLDH, and/or AST, and <3-fold in GGT and/or ALP	Hepatocellular predominance (acute necrosis, ischemic or toxic damage to hepatocyte)
>5–10-fold increase in GGT and/or ALP, and <3-fold in SDH, GLDH and/or AST	Biliary predominance (cholestasis, cholangitis, choletithus)
>10-fold increase in SDH, GLDH and/or AST and GGT and or ALP	Mixed liver injury (acute hepatocellular and biliary damage)

**Table 5 animals-12-00861-t005:** Enzymatic profiles of different diseases documented in equine clinical pathology.

		Increase in Enzymatic Activity Related to Reference Value Intervals	
Causes	Pathology	GLDH	SDH	AST	GGT	ALP	LDH	References
Infectious	Theiler’s disease	 	 	  	 			Divers [3]
Theiler’s disease-associated to EqPV-H			 	 			Tomlinson et al. [16]; Vengust et al. [50]
Nonprimate hepacivirus (NPHV)							Gather et al. [7]; Lyons et al. [51]; Ramsay et al. [42]
Equine pegivirus (EPgV)							Figueiredo et al. [14]; Ramsay et al. [42]
Equine parvovirus			 	  	 		Ramsauer et al. [15]
Tyzzer’s disease		 		  			Haggett et al. [32]
Inflammatory	Cholangiohepatitis				  	 		Peek [52]
Acute biliary obstruction				  			Gardner et al. [26]; Divers [3]
Colic		 		 			Saules et al. [53]; Underwood et al. [54]
Cholelithiasis				  	 		Ryu et al. [55]
Liver lobe torsion							Tennent-Brown et al. [56]
Cholangiohepatitis and Choledocholithiasis				 			Mair and Love [57]
Chronic active hepatitis (CAH)				  			Mair and Love [57]
Cholelithiasis			 	  	  		Hoffmann and Solter [19]; Peek and Divers [58]
Biliary atresia		  		  	 		van der Luer and Kroneman [59]
Proximal enteritis			  	  	  		Davis et al. [31]
Intoxication	Plants: clover, panicum grasses		 	 	 	 		Elfenbein and House [60]
Mycotoxins			 	  			Braga et al. [61]; Dänicke et al. [62]
Copper			 	  	 	 	Ankringa et al. [29]
Iron			  	  	 	 	Theelen et al. [63]
Pyrrolizidine alkaloid toxicity				  	 		Curran et al. [46]; Mair and Love [57]
Neoplasia	Primary liver tumors	  			  	 		Beeler-Marfisi et al. [18]
Metastatic infiltration of the liver	  			  			West [2]
Other situations	Amyloidosis				  	  		Abdelkader et al. [64]; Hablolvarid et al. [65]
Hyperammonemia		 	 				McConnico et al. [66]
Hepatic lipidosis		 		 			West [2]
Glycogen branching enzyme deficiency			 	 			Valberg and Mickelson [67]
Hyperlipemia			 	 			Seifi et al. [68]; Dunkel et al. [69]; Oikawa et al. [70]
Hepatic abscess							Sellon et al. [71]

Arrows indicate:

, mild elevation; 



, moderate elevation; and 





, marked elevation of different enzymes.

## Data Availability

No new data were created or analyzed in this study. Data sharing is not applicable to this article.

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
