# Peer review of "Hepatic Enzyme Profile in Horses"

_animals, 2022, doi:10.3390/ani12070861_

Round 1

Reviewer 1 Report

The manuscript is well written and clear. Nevertheless, although focused on hepatic enzymatic, in my opinion the manuscript could eventually  also discuss in deeper detail their diagnostic roles in correlation with other parameters such as direct / indirect bilirubin, bile acids, albumin, globulins, total proteins, glucose, coagulation factors ( PT, aPTT, fibrinogen) etc.

Author Response

Dear reviewer,

Thank you very much for your comments. We are pleased that "Animals" allowed us to review our manuscript "Hepatic clinical enzyme profile in horses" (Manuscript ID: animals-1647503) before publication. We also appreciate your constructive comments and suggestions, which improved this work. In the revised manuscript, we have addressed the comments and modifications suggested by you. Below, we present the detailed response to the comments. For ease of review, all significant changes in the revised manuscript have been highlighted in red.

Response to Reviewer 1

The manuscript is well written and clear. Nevertheless, although focused on hepatic enzymatic, in my opinion the manuscript could eventually  also discuss in deeper detail their diagnostic roles in correlation with other parameters such as direct / indirect bilirubin, bile acids, albumin, globulins, total proteins, glucose, coagulation factors ( PT, aPTT, fibrinogen) etc.

Response:

Dear reviewer,

We appreciate your constructive comments and suggestions on our manuscript "Hepatic clinical enzyme profile in horses" (Manuscript ID: animals-1647503). We would also like to emphasize that this review was carefully prepared based on the available literature and we have focused specifically on enzymes because of their importance in the diagnosis of hepatobiliary pathology in the equine. It would have been interesting to incorporate the rest of the parameters you mention. However, the authors have wanted to make a focused, detailed, and concrete study of the pathologies and diagnosis based on hepatobiliary enzymes, when different hepatic and/or biliary alterations occur in the horse.

The authors would like to write a similar article on the interpretation of these parameters to which you refer in the near future.

Sincerely

Reviewer 2 Report

The manuscript describes thoroughly the relation between several enzymes and liver diseases in horses. I think it is a well-written and informative review.

Author Response

Dear reviewer,

Thank you very much for your comments. We are pleased that "Animals" allowed us to review our manuscript "Hepatic clinical enzyme profile in horses" (Manuscript ID: animals-1647503) before publication. We also appreciate your constructive comments and suggestions, which improved this work. In the revised manuscript, we have addressed the comments and modifications suggested by you. Below, we present the detailed response to the comments. For ease of review, all significant changes in the revised manuscript have been highlighted in red.

Response to Reviewer 2

The manuscript describes thoroughly the relation between several enzymes and liver diseases in horses. I think it is a well-written and informative review.

Response:

Dear reviewer,

Thanks for your comments and time when reviewing this manuscript. We are also thankful for your positive and constructive comments on the review.

Reviewer 3 Report

In humans….Although GGT is a less specific marker of liver function, higher GGT levels have been found to be an independent predictor of the incidence of type 2 diabetes…as evident in…The association between liver enzymes and risk of type 2 diabetes: the Namwon study. Diabetol Metab Syndr 6, 14 (2014). https://doi.org/10.1186/1758-5996-6-14……        and what about the finding that  acute steatosis is possibly diagnosed by liver enzymes?

Authors could discuss these intriguing points.

Author Response

Dear reviewer,

Thank you very much for your comments. We are pleased that "Animals" allowed us to review our manuscript "Hepatic clinical enzyme profile in horses" (Manuscript ID: animals-1647503) before publication. We also appreciate your constructive comments and suggestions, which improved this work. In the revised manuscript, we have addressed the comments and modifications suggested by you. Below, we present the detailed response to the comments. For ease of review, all significant changes in the revised manuscript have been highlighted in red.

Response to Reviewer 3

In humans….Although GGT is a less specific marker of liver function, higher GGT levels have been found to be an independent predictor of the incidence of type 2 diabetes…as evident in…The association between liver enzymes and risk of type 2 diabetes: the Namwon study. Diabetol Metab Syndr 6, 14 (2014). https://doi.org/10.1186/1758-5996-6-14……        and what about the finding that  acute steatosis is possibly diagnosed by liver enzymes?

 Authors could discuss these intriguing points.

Response:

Dear reviewer,

The association between liver enzymes and risk of type 2 diabetes in humans as well as other endocrine conditions related with hyperlipidemia and therefore with the risk of hepatic lipidosis have been added, as suggested by the reviewer. The authors thank the reviewer for this suggestion when introducing these interesting contents.

“Whereas GGT is a less specific marker of liver alteration in humans, it has been shown that high levels of GGT can be an independent predictor of the incidence of type 2 diabetes [36]. In horses, type 2 diabetes mellitus is often associated with pituitary pars intermedia dysfunction (PPID) [37]. Indeed, different alterations in lipid metabolism such as lipolysis or ketogenesis have been detailed in horses with PPID [38]. In ponies, cases of persistent hypertriglycemia, hyperinsulinemia, and hyperglycemia attributed to PPID have been reported, as well as an increase in the activity of different liver enzymes such as ALP with values up to 204 IU/L, AST (693 IU/L), GGT (46 IU/L) and SDH (10 IU/L). Likewise, liver biopsies have shown important changes in the vacuoles of liver cells corresponding to hepatic lipidosis [39].

References:

  1. Ahn, H.R., Shin, M.H., Nam, H.S. Park, K.S., Lee, YH, Jeong, S.K., Choi, J.S., Kweon, S.S. The association between liver enzymes and risk of type 2 diabetes: the Namwon study. Diabetol. Metab. Syndr.2014, 6,14. 
  2. Durham, A.E., Hughes, K.J., Cottle, H.J., Rendle, D.I.and Boston, R.C.(2009) Type 2 diabetes mellitus with pancreatic beta cell dysfunction in 3 horses confirmed with minimal model analysis. Equine Vet. J. 41, 924- 929.
  3. Frank, N., Elliott, S.B., Brandt, L.E. Keisler, D.H.Physical characteristics, blood hormone concentrations, and plasma lipid concentrations in obese horses with insulin resistance. J. Am. Vet. Med. Assoc. 2006, 228, 1383- 1390.
  4. Dunkel, B., Wilford, S.A., Parkinson, N.J., Ward, C., Smith, P., Grahame, L., Brazil, T., Schott, H.C. 2nd. Severe hypertriglyceridaemia in horses and ponies with endocrine disorders. Equine Vet. J. 2014, 46, 118-22. 

Sincerely
